# SHORT AND SPARSE DECONVOLUTION — A GEOMETRIC APPROACH

**Yenson Lau**[*]
Electrical Engineering
Columbia University
y.lau@columbia.edu

**Qing Qu**[*]
Center for Data Science
New York University
qq213@nyu.edu

**Han-wen Kuo**
Electrical Engineering
Columbia University
hk2673@columbia.edu

**Pengcheng Zhou**
Department of Statistics
Columbia University
zhoupc2018@gmail.com

**Yuqian Zhang**
Electrical & Computer Engineering
Rutgers University
yqz.zhang@rutgers.edu

**John Wright**
Electrical Engineering
Columbia University
jw2966@columbia.edu

## ABSTRACT

Short-and-sparse deconvolution (SaSD) is the problem of extracting localized, recurring motifs in signals with spatial or temporal structure. Variants of this problem arise in applications such as image deblurring, microscopy, neural spike sorting, and more. The problem is challenging in both theory and practice, as natural optimization formulations are nonconvex. Moreover, practical deconvolution problems involve smooth motifs (kernels) whose spectra decay rapidly, resulting in poor conditioning and numerical challenges. This paper is motivated by recent theoretical advances (Zhang et al., 2017; Kuo et al., 2019), which characterize the optimization landscape of a particular nonconvex formulation of SaSD and give a *provable* algorithm which exactly solves certain non-practical instances of the SaSD problem. We leverage the key ideas from this theory (sphere constraints, data-driven initialization) to develop a *practical* algorithm, which performs well on data arising from a range of application areas. We highlight key additional challenges posed by the ill-conditioning of real SaSD problems, and suggest heuristics (acceleration, continuation, reweighting) to mitigate them. Experiments demonstrate the performance and generality of the proposed method.

## 1 INTRODUCTION

Many signals arising in science and engineering can be modeled as superpositions of basic, recurring motifs, which encode critical information about a physical process of interest. Signals of this type can be modeled as the convolution of a zero-padded short kernel $\boldsymbol{a}_0 \in \mathbb{R}^{p_0}$ (the motif) with a longer sparse signal $\boldsymbol{x}_0 \in \mathbb{R}^m$ ($m \gg p_0$) which encodes the locations of the motifs in the sample[1]:

$$\boldsymbol{y} \;=\; \iota \boldsymbol{a}_0 \;\circledast\; \boldsymbol{x}_0. \tag{1}$$

We term this a short-and-sparse (SaS) model. Since often only $\boldsymbol{y}$ is observed, *short-and-sparse deconvolution* (SaSD) is the problem of recovering both $\boldsymbol{a}_0$ and $\boldsymbol{x}_0$ from $\boldsymbol{y}$. Variants of SaSD arise in areas such as microscopy (Cheung et al., 2018), astronomy (Briers et al., 2013), and neuroscience (Song et al., 2018). SaSD is a challenging inverse problem in both theory and practice. Natural formulations are nonconvex, and very little algorithmic theory was available. Moreover, practical instances are often ill-conditioned, due to the spectral decay of the kernel $\boldsymbol{a}_0$ (Cheung et al., 2018).

This paper is motivated by recent theoretical advances in nonconvex optimization and, in particular, on the geometry of SaSD. Zhang et al. (2017) and Kuo et al. (2019) study particular optimization

---

[*]YL and QQ contributed equally to this work. The full version of this work can be found at `https://arxiv.org/abs/1908.10959`.

[1]For simplicity, (1) uses cyclic convolution; algorithms are results also apply to linear convolution with minor modifications. Here $\iota$ denotes the zero padding operator.

formulations for SaSD and show that the landscape is largely driven by the *problem symmetries* of SaSD. They derive provable methods for idealized problem instances, which exactly recover $(\boldsymbol{a}_0, \boldsymbol{x}_0)$ up to trivial ambiguities. While inspiring, these methods are *not practical* and perform poorly on real problem instances. Where the emphasis of Zhang et al. (2017) and Kuo et al. (2019) is on theoretical guarantees, here we focus on practical computation. We show how to combine ideas from this theory with heuristics that better address the properties of practical deconvolution problems, to build a novel method that performs well on data arising in a range of application areas. A critical issue in moving from theory to practice is the poor conditioning of naturally-occurring deconvolution problems: we show how to address this with a combination of ideas from sparse optimization, such as momentum, continuation, and reweighting. The end result is a general purpose method, which we demonstrate on data from neural spike sorting, calcium imaging and fluorescence microscopy.

**Notation.** The zero-padding operator is denoted by $\iota : \mathbb{R}^p \to \mathbb{R}^m$. Projection of a vector $\boldsymbol{v} \in \mathbb{R}^p$ onto the sphere is denoted by $\mathcal{P}_{\mathbb{S}^{p-1}}(\boldsymbol{v}) \doteq \boldsymbol{v} / \|\boldsymbol{v}\|_2$, and $\mathcal{P}_{\boldsymbol{z}}(\boldsymbol{v}) \doteq \boldsymbol{v} - \langle \boldsymbol{v}, \boldsymbol{z} \rangle \boldsymbol{z}$ denotes projection onto the tangent space of $\boldsymbol{z} \in \mathbb{S}^{p-1}$. The Riemannian gradient of a function $f : \mathbb{S}^{p-1} \to \mathbb{R}$ at point $\boldsymbol{z}$ on the sphere is given by $\operatorname{grad} f(\boldsymbol{z}) \doteq \mathcal{P}_{\boldsymbol{z}}(\nabla f(\boldsymbol{z}))$.

**Reproducible research.** The code for implementations of our algorithms can be found online:



[https://github.com/qingqu06/sparse_deconvolution](https://github.com/qingqu06/sparse_deconvolution).



For more details of our work on SaSD, we refer interested readers to our project website



[https://deconvlab.github.io/](https://deconvlab.github.io/).



## 2  SYMMETRY AND GEOMETRY IN SASD

In this section, we begin by describing two intrinsic properties for SaSD. Later, we show how these play an important role in the geometry of optimization and the design of efficient methods.

An important observation of the SaSD problem is that it admits multiple equivalent solutions. This is purely due to the cyclic convolution between $\boldsymbol{a}_0$ and $\boldsymbol{x}_0$, which exhibits the trivial ambiguity[2]

$$\boldsymbol{y} \;=\; \iota \boldsymbol{a}_0 \circledast \boldsymbol{x}_0 \;=\; (\alpha s_\ell [\iota \boldsymbol{a}_0]) \;\circledast\; \left(\tfrac{1}{\alpha} s_{-\ell} [\boldsymbol{x}_0]\right),$$

for any nonzero scalar $\alpha$ and cyclic shift $s_\ell [\cdot]$. These scale and shift symmetries create several acceptable candidates for $\boldsymbol{a}_0$ and $\boldsymbol{x}_0$, and in the absence of further information we only expect to recover $\boldsymbol{a}_0$ and $\boldsymbol{x}_0$ up to symmetry. Furthermore, they largely drive the behavior of certain nonconvex optimization problems formulated for SaSD. Since the success of SaSD requires distinguishing between overlapping copies of $\boldsymbol{a}_0$, its difficulty also depends highly on the "similarity" of the $\boldsymbol{a}_0$ to its shifts. Here we capture this notion using the *shift-coherence* of $\boldsymbol{a}_0$,

$$\mu(\boldsymbol{a}_0) \doteq \max_{\ell \neq 0} |\langle \iota \boldsymbol{a}_0, s_\ell [\iota \boldsymbol{a}_0] \rangle| \;\in\; [0, 1] . \tag{2}$$

Intuitively, the shifts of $\boldsymbol{a}_0$ become closer together as $\mu(\boldsymbol{a}_0)$ increases (Figure 10), making objective landscapes for optimization less favorable for recovering any specific shift of $\boldsymbol{a}_0$.

### 2.1  LANDSCAPE GEOMETRY UNDER SHIFT-INCOHERENCE

A natural approach to solving SaSD is to formulate it as a suitable optimization problem. In this paper we will focus on the *Bilinear Lasso* (BL) problem, which minimizes the squared error between the observation $\boldsymbol{y}$ and its reconstruction $\boldsymbol{a} \circledast \boldsymbol{x}$, plus a $\ell_1$-norm sparsity penalty on $\boldsymbol{x}$,

$$\min_{\boldsymbol{a} \in \mathbb{S}^{p-1}, \boldsymbol{x} \in \mathbb{R}^m} \left[ \Psi_{\mathrm{BL}}(\boldsymbol{a}, \boldsymbol{x}) \;\doteq\; \tfrac{1}{2} \|\boldsymbol{y} - \iota \boldsymbol{a} \circledast \boldsymbol{x}\|_2^2 + \lambda \|\boldsymbol{x}\|_1 \right] . \tag{3}$$

Later in this section, we will see that the kernel length $p$ should be set slightly larger than $p_0$.

The Bilinear Lasso is a nonconvex optimization problem, as the shift symmetries of SaSD create discrete local minimizers in the objective landscape. The regularization created by problem symmetries

---

[2]We therefore assume w.l.o.g. that $\|\boldsymbol{a}_0\|_2 = 1$ in this paper.

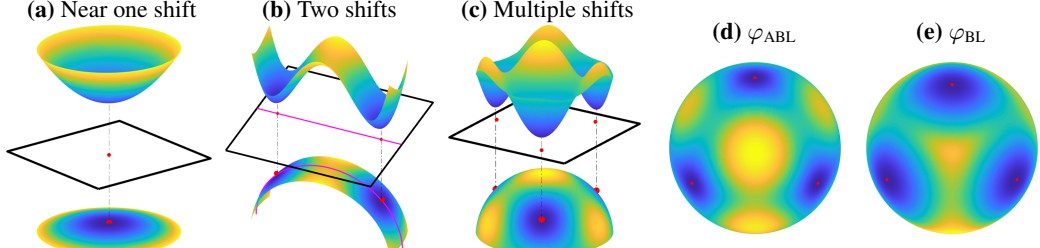

Figure 1: **Geometry of $\varphi_{\mathrm{ABL}}$ near superpositions of shifts** of $\boldsymbol{a}_0$ (Kuo et al., 2019). **(a)** Regions near single shifts are strongly convex. **(b)** Regions between two shifts contain a saddle-point, with negative curvature towards each shift and positive curvature orthogonally. **(c)** The span of three shifts. For each figure, the top shows the function value in height, and the bottom shows function value over the sphere. **(d,e)** When $\mu_s(\boldsymbol{a}_0) \approx 0$, the Bilinear Lasso $\varphi_{\mathrm{BL}}(\boldsymbol{a}) \doteq \min_{\boldsymbol{x}} \Psi_{\mathrm{BL}}(\boldsymbol{a}, \boldsymbol{x})$ and ABL $\varphi_{\mathrm{ABL}}(\boldsymbol{a})$ are empirically similar in the span of three shifts.

in nonconvex inverse problems are a fairly general phenomenon (Sun et al., 2015) and, as Kuo et al. (2019) shows, its influence in SaSD extends beyond the neighborhoods of these local minimizers. Kuo et al. analyzed an *Approximate Bilinear Lasso* (ABL) objective[3] $\Psi_{\mathrm{ABL}}$, which satisfies

$$\Psi_{\mathrm{ABL}}(\boldsymbol{a}, \boldsymbol{x}) \simeq \Psi_{\mathrm{BL}}(\boldsymbol{a}, \boldsymbol{x}), \qquad \text{when } \mu(\boldsymbol{a}) \simeq 0.$$

This non-practical objective serves as a valid simplification of the Bilinear Lasso for analysis when the true kernel is itself incoherent, i.e. $\mu(\boldsymbol{a}_0) \simeq 0$ (Figures 1d and 1e). Under its marginalization[4]

$$\varphi_{\mathrm{ABL}}(\boldsymbol{a}) \doteq \min_{\boldsymbol{x} \in \mathbb{R}^m} \Psi_{\mathrm{ABL}}(\boldsymbol{a}, \boldsymbol{x}), \tag{4}$$

certain crucial properties regarding its curvature can be characterized for generic choices of $\boldsymbol{x}$. The reason we choose to partial minimize $\boldsymbol{x}$ instead of $\boldsymbol{a}$ is because *(i)* the problem (4) is convex w.r.t. $\boldsymbol{x}$, and *(ii)* the dimension of the subspace of $\boldsymbol{a}$ is significantly smaller than that of $\boldsymbol{x}$ (i.e., $p \ll m$), which is the place that the measure concentrates.

**Curvature in the span of a few shifts.** Suppose we set $p > p_0$, which ensures that we can find an $\boldsymbol{a} \simeq \alpha_1 s_{\ell_1}[\boldsymbol{a}_0] + \alpha_2 s_{\ell_2}[\boldsymbol{a}_0] \in \mathbb{S}^{p-1}$ that lies near the span of two shifts of $\boldsymbol{a}_0$. If $\alpha_1 \simeq \pm 1$ (or $\alpha_2 \simeq 0$) then, under suitable conditions on $\boldsymbol{a}_0$ and $\boldsymbol{x}_0$, Kuo et al. (2019) asserts that $\boldsymbol{a}$ lies in a strongly convex region of $\varphi_{\mathrm{ABL}}$, containing a single minimizer near $s_{\ell_1}[\boldsymbol{a}_0]$ (Figure 1a); the converse is also true. A saddle-point exists nearby when $\alpha_1 \simeq \alpha_2$ is balanced, characterized by large negative curvature along the two shifts and positive curvature in orthogonal directions (Figure 1b). Interpolating between these two cases, large negative gradients point towards individual shifts.

The behavior of $\varphi_{\mathrm{ABL}}$ between two shifts of $\boldsymbol{a}_0$ — strong convexity near single shifts, and saddle-points near balanced points — extends to regions of the sphere spanned by several shifts (Figure 1c); we elaborate on this further in Appendix A.1. This regional landscape guarantees that $\boldsymbol{a}_0$ can be efficiently recovered up to a signed shift using methods for first and second-order descent, as soon as $\boldsymbol{a}$ can be brought sufficiently close to the span of a few shifts.

**Optimization over the sphere.** For both the Bilinear Lasso and ABL, a unit-norm constraint on $\boldsymbol{a}$ is enforced to break the scaling symmetry between $\boldsymbol{a}_0$ and $\boldsymbol{x}_0$. Choosing the $\ell_2$-norm, however, has surprisingly strong implications for optimization. The ABL objective, for example, is piecewise concave whenever $\boldsymbol{a}$ is sufficiently far away from any shift of $\boldsymbol{a}_0$, but the sphere induces positive curvature near individual shifts to create strong convexity. These two properties combine to ensure recoverability of $\boldsymbol{a}_0$. In contrast, enforcing $\ell_1$-norm constraints often leads to spurious minimizers for deconvolution problems (Levin et al., 2011; Benichoux et al., 2013; Zhang et al., 2017).

**Initializing near a few shifts.** The landscape of $\varphi_{\mathrm{ABL}}$ makes single shifts of $\boldsymbol{a}_0$ easy to locate if $\boldsymbol{a}$ is initialized near a span of a few shifts. Fortunately, this is a relatively simple matter in SaSD, as $\boldsymbol{y}$ *is*

---

[3]As the intention here is apply some key intuition from the ABL objective towards the Bilinear Lasso itself, we intentionally omit the concrete form of $\Psi_{\mathrm{ABL}}(\boldsymbol{a})$. Readers may refer to Appendix A for more details.

[4]Minimizing $\varphi_{\mathrm{ABL}}$, this is equivalent to minimizing $\Psi_{\mathrm{ABL}}$ as $\boldsymbol{x}$ can be recovered via convex optimization.

*itself a sparse superposition of shifts.* Therefore, one initialization strategy is to randomly choose a length-$p_0$ window $\widetilde{\boldsymbol{y}}_i \doteq [y_i\ y_{i+1} \ldots y_{i+p_0-1}]^T$ from the observation and set

$$\boldsymbol{a}^{(0)} \doteq \mathcal{P}_{\mathbb{S}^{p-1}}\big(\big[\ \boldsymbol{0}_{p_0-1}\ ;\ \widetilde{\boldsymbol{y}}_i\ ;\ \boldsymbol{0}_{p_0-1}\ \big]\big). \tag{5}$$

This brings $\boldsymbol{a}^{(0)}$ suitably close to the sum of a few shifts of $\boldsymbol{a}_0$ (Appendix A.2); any truncation effects are absorbed by padding the ends of $\widetilde{\boldsymbol{y}}_i$, which also sets the length for $\boldsymbol{a}$ to be $p = 3p_0 - 2$.

**Implications for practical computation.** The (regionally) benign optimization landscape of $\varphi_{\mathrm{ABL}}$ guarantees that efficient recovery is possible for SaSD when $\boldsymbol{a}_0$ is incoherent. Applications of sparse deconvolution, however, are often motivated by sharpening or resolution tasks (Huang et al., 2009; Candès & Fernandez-Granda, 2014; Campisi & Egiazarian, 2016) where the motif $\boldsymbol{a}_0$ is smooth and coherent (i.e. $\mu(\boldsymbol{a}_0)$ is large). The ABL objective is a poor approximation of the Bilinear Lasso in such cases and fails to yield practical algorithms, so we should optimize the Bilinear Lasso directly.

From Figures 1d and 1e, we can see that low-dimensional subspheres spanned by shifts of $\boldsymbol{a}_0$ are empirically similar when $\boldsymbol{a}_0$ is incoherent. Although this breaks down in the coherent case, as we illustrate in Appendix A.3, the symmetry breaking properties of $\varphi_{\mathrm{BL}}$ remain present. This allows us to apply the geometric intuition discussed here to create an optimization method that, with the help of a number of computational heuristics, performs well in for SaSD even in general problem instances.

---

**Algorithm 1** Inertial Alternating Descent Method (iADM)

---

**Input:** Initializations $\boldsymbol{a}^{(0)} \in \mathbb{S}^{p-1}$, $\boldsymbol{x} \in \mathbb{R}^m$; observation $\boldsymbol{y} \in \mathbb{R}^m$; penalty $\lambda \geqslant 0$; momentum $\alpha \in [0,1)$.
**Output:** $(\boldsymbol{a}^{(k)}, \boldsymbol{x}^{(k)})$, a local minimizer of $\Psi_{\mathrm{BL}}$.
   *Initialize* $\boldsymbol{a}^{(1)} = \boldsymbol{a}^{(0)}$, $\boldsymbol{x}^{(1)} = \boldsymbol{x}^{(0)}$.
   **for** $k = 1, 2, \ldots$ until converged **do**
      *Update $\boldsymbol{x}$* with accelerated proximal gradient step:

$$\boldsymbol{w}^{(k)} \ \leftarrow\ \boldsymbol{x}^{(k)} + \alpha \cdot \big(\boldsymbol{x}^{(k)} - \boldsymbol{x}^{(k-1)}\big)$$

$$\boldsymbol{x}^{(k+1)} \ \leftarrow\ \mathrm{soft}_{\lambda t_k}\big[\boldsymbol{w}^{(k)} - t_k \cdot \nabla_{\boldsymbol{x}} \psi_\lambda\big(\boldsymbol{a}^{(k)}, \boldsymbol{w}^{(k)}\big)\big],$$

     where $\mathrm{soft}_\lambda(\boldsymbol{v}) \doteq \mathrm{sign}(\boldsymbol{v}) \odot \max(|\boldsymbol{v} - \lambda|, \boldsymbol{0})$ denotes the soft-thresholding operator.
      *Update $\boldsymbol{a}$* with accelerated Riemannian gradient step:

$$\boldsymbol{z}^{(k)} \ \leftarrow\ \mathcal{P}_{\mathbb{S}^{p-1}}\big(\boldsymbol{a}^{(k)} + \tfrac{\alpha}{\langle \boldsymbol{a}^{(k)}, \boldsymbol{a}^{(k-1)}\rangle} \cdot \mathcal{P}_{\boldsymbol{a}^{(k-1)}}\big(\boldsymbol{a}^{(k)}\big)\big)$$

$$\boldsymbol{a}^{(k+1)} \ \leftarrow\ \mathcal{P}_{\mathbb{S}^{p-1}}\big(\boldsymbol{z}^{(k)} - \tau_k \cdot \mathrm{grad}_{\boldsymbol{a}}\, \psi_\lambda\big(\boldsymbol{z}^{(k)}, \boldsymbol{x}^{(k+1)}\big)\big).$$

   **end for**

---

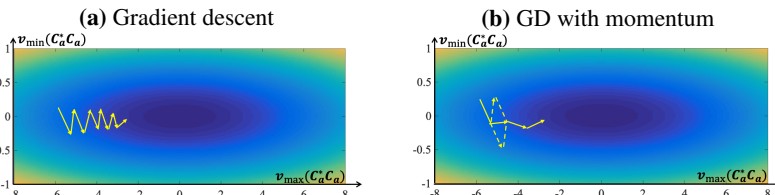

**(a)** Gradient descent        **(b)** GD with momentum

Figure 2: **Momentum acceleration.** a) Iterates of gradient descent oscillate on ill-conditioned functions; each marker denotes one iteration. b) Momentum dampens oscillation and speeds up convergence.

## 3   Designing a practical SaSD algorithm

Several algorithms for SaSD-type problems have been developed for specific applications, such as image deblurring (Levin et al., 2011; Briers et al., 2013; Campisi & Egiazarian, 2016), neuroscience (Rey et al., 2015; Friedrich et al., 2017; Song et al., 2018), and image super-resolution (Baker & Kanade, 2002; Shtengel et al., 2009; Yang et al., 2010), or are augmented with additional structure (Wipf & Zhang, 2014; Ling & Strohmer, 2017; Walk et al., 2017).

Here, we instead leverage the theory from Section 2 to build an algorithm for general practical settings. In addition to applying an appropriate initialization scheme (5) and optimizing on the sphere, we minimize the Bilinear Lasso (3) instead of the ABL (4) to more accurately account for interactions between shifts of $\boldsymbol{a}_0$ in highly shift-coherent settings. Furthermore, we also address the negative effects of large coherence using a number of heuristics, leading to an efficient algorithm for SaSD.

**Momentum acceleration.** In shift-coherent settings, the Hessian of $\Psi_{\mathrm{BL}}$ becomes ill-conditioned[5] near shifts of $\boldsymbol{a}_0$, a situation known to cause slow convergence for first-order methods (Nesterov, 2013). A remedy is to add momentum (Polyak, 1964; Beck & Teboulle, 2009) to first-order iterations, for instance, by augmenting gradient descent on some smooth $f(\boldsymbol{z})$ with stepsize $\tau$ with the term $\boldsymbol{w}$,

$$\boldsymbol{w}^{(k)} \leftarrow \boldsymbol{z}^{(k)} + \alpha \cdot (\boldsymbol{z}^{(k)} - \boldsymbol{z}^{(k-1)}) \tag{6}$$

$$\boldsymbol{z}^{(k+1)} \leftarrow \boldsymbol{w}^{(k)} - \tau \cdot \nabla f(\boldsymbol{w}^{(k)}). \tag{7}$$

Here, $\alpha$ controls the momentum added[6]. As illustrated in Figure 2, this additional term improves convergence by reducing oscillations of the iterates for ill-conditioned problems. Momentum has been shown to improve convergence for nonconvex and nonsmooth problems (Pock & Sabach, 2016; Jin et al., 2018). Here we provide an inertial alternating descent method (iADM) for finding local minimizers of $\Psi_{\mathrm{BL}}$ (Algorithm 1), which modifies iPALM (Pock & Sabach, 2016) to perform updates on $\boldsymbol{a}$ via retraction on the sphere (Absil et al., 2009)[7].

---

**Algorithm 2** SaS-BD with homotopy continuation

---

**Input:** Observation $\boldsymbol{y} \in \mathbb{R}^m$, motif size $p_0$; momentum $\alpha \in [0, 1)$; initial $\lambda^{(1)}$ final $\lambda^\star$, penalty decrease $\eta \in (0, 1)$; precision factor $\delta \in (0, 1)$.
**Output:** Solution path $\{(\hat{\boldsymbol{a}}^{(n)}, \hat{\boldsymbol{x}}^{(n)}; \lambda^{(n)})\}$ for SaSD.
   *Set* number of iterations $N \leftarrow \lfloor \log(\lambda^\star/\lambda^{(1)}) / \log \eta \rfloor$.
   *Initialize* $\hat{\boldsymbol{a}}^{(0)} \in \mathbb{R}^{3p_0 - 2}$ using (5), $\hat{\boldsymbol{x}}^{(0)} = \boldsymbol{0} \in \mathbb{R}^m$.
   **for** $n = 1, \ldots, N$ **do**
      *Minimize* $\Psi_{\lambda^{(n)}}$ to precision $\delta\lambda^{(n)}$ with Algorithm 1:
$$\left(\hat{\boldsymbol{a}}^{(n)}, \hat{\boldsymbol{x}}^{(n)}\right) \leftarrow \mathrm{iADM}\left(\hat{\boldsymbol{a}}^{(n-1)}, \hat{\boldsymbol{x}}^{(n-1)}; \boldsymbol{y}, \lambda^{(n)}, \alpha\right).$$
      *Update* $\lambda^{(n+1)} \leftarrow \eta\lambda^{(n)}$.
   **end for**

---

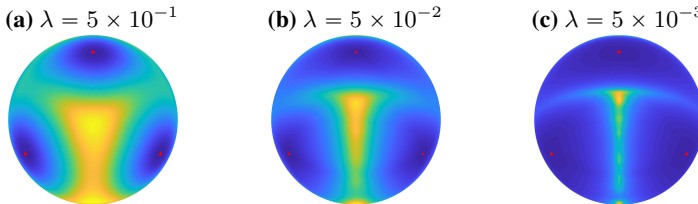

**(a)** $\lambda = 5 \times 10^{-1}$     **(b)** $\lambda = 5 \times 10^{-2}$     **(c)** $\lambda = 5 \times 10^{-3}$

Figure 3: **Bilinear-lasso objective $\varphi_\lambda$ on the sphere** $\mathbb{S}^{p-1}$, for $p = 3$ and varying $\lambda$; brighter colors indicate higher values. The function landscape of $\varphi_\lambda$ flattens as sparse penalty $\lambda$ decreases from left to right.

**Homotopy continuation.** It is also possible to improve optimization by modifying the objective $\Psi_{\mathrm{BL}}$ directly through the sparsity penalty $\lambda$. Variations of this idea appear in both Zhang et al. (2017) and Kuo et al. (2019), and can also help to mitigate the effects of large shift-coherence.

When solving (3) in the noise-free case, it is clear that larger choices of $\lambda$ encourage sparser solutions for $\boldsymbol{x}$. Conversely, smaller choices of $\lambda$ place local minimizers of the marginal objective $\varphi_{\mathrm{BL}}(\boldsymbol{a}) \doteq \min_{\boldsymbol{x}} \Psi_{\mathrm{BL}}(\boldsymbol{a}, \boldsymbol{x})$ closer to signed-shifts of $\boldsymbol{a}_0$ by emphasizing reconstruction quality. When $\mu(\boldsymbol{a}_0)$ is large, however, $\varphi_{\mathrm{BL}}$ becomes ill-conditioned as $\lambda \to 0$ due to the poor spectral conditioning of $\boldsymbol{a}_0$, leading to severe flatness near local minimizers and the creation spurious local minimizers when noise is present (Figure 3). Conversely, larger values of $\lambda$ limit $\boldsymbol{x}$ to a small set of support patterns and simplify the landscape of $\varphi_{\mathrm{BL}}$, at the expense of precision.

It is therefore important both for fast convergence and accurate recovery for $\lambda$ to be chosen appropriately. When problem parameters — such as noise level, $p_0$, or $\theta$ — are not known a priori, a *homotopy continuation method* (Hale et al., 2008; Wright et al., 2009; Xiao & Zhang, 2013) can be used to obtain a range of solutions for SaSD. Using initialization (5), a rough estimate $(\hat{\boldsymbol{a}}^{(1)}, \hat{\boldsymbol{x}}^{(1)})$

---

[5]This is because the circulant matrix $\boldsymbol{C}_{\boldsymbol{a}_0}$ is ill-conditioned.

[6]Setting $\alpha = 0$ removes momentum and reverts to standard gradient descent.

[7]The stepsizes $t_k$ and $\tau_k$ are obtained by backtracking (Nocedal & Wright, 2006; Pock & Sabach, 2016) to ensure sufficient decrease for $\Psi_{\mathrm{BL}}(\boldsymbol{a}^{(k)}, \boldsymbol{w}^{(k)}) - \Psi_{\mathrm{BL}}(\boldsymbol{a}^{(k)}, \boldsymbol{x}^{(k+1)})$, and vice versa.

is obtained by solving (3) with iADM using a large choice for $\lambda^{(1)}$. This estimate is refined via a *solution path* $\{(\hat{a}^{(n)}, \hat{x}^{(n)}; \lambda^{(n)})\}$ by gradually decreasing $\lambda^{(n)}$. By ensuring that $x$ remains sparse along the solution path, the objective $\Psi_{\mathrm{BL}}$ enjoys restricted strong convexity w.r.t. both $a$ and $x$ throughout optimization (Agarwal et al., 2010). As a result, homotopy achieves linear convergence for SaSD where sublinear convergence is expected otherwise (Figures 4c and 4d). We provide a complete algorithm for SaSD combining Bilinear Lasso and homotopy continuation in Algorithm 2.

## 4 EXPERIMENTS

### 4.1 SYNTHETIC EXPERIMENTS

Here we perform SaSD in simulations on both coherent and incoherent settings. Coherent kernels are discretized from the Gaussian window function $a_0 = g_{p_0,0.5}$, where $g_{p,\sigma} \doteq \mathcal{P}_{\mathbb{S}^{p-1}}\left(\left[\exp\left(-\frac{(2i-p-1)^2}{\sigma^2(p-1)^2}\right)\right]_{i=1}^{p}\right)$. Incoherent kernels $a_0 \sim \mathrm{Unif}(\mathbb{S}^{p_0-1})$ are sampled uniformly on the sphere.

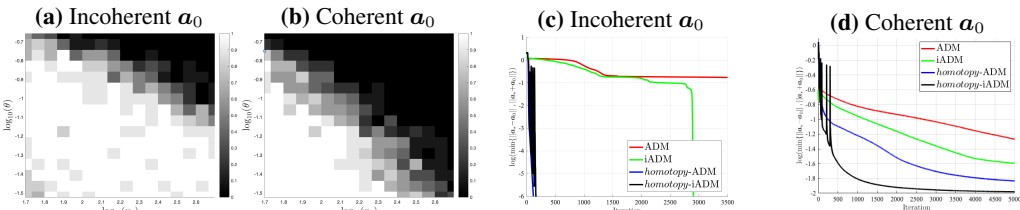

**(a)** Incoherent $a_0$    **(b)** Coherent $a_0$    **(c)** Incoherent $a_0$    **(d)** Coherent $a_0$

Figure 4: **Synthetic experiments for Bilinear Lasso.** *Success probability* **(a, b)**: $x_0 \sim_{\mathrm{i.i.d.}} \mathcal{BR}(\theta)$, the success probability of SaS-BD by solving (3), shown by increasing brightness, is large when the sparsity rate $\theta$ is sufficiently small compared to the length of $a_0$, and vice versa. Success with a fixed sparsity rate is more likely when $a_0$ is incoherent. *Algorithmic convergence* **(c, d)**: iterate convergence for iADM with $\alpha_k = (k-1)/(k+1)$ vs. $\alpha_k = 0$ (ADM); with and without homotopy. Homotopy significantly improves convergence rate, and momentum improves convergence when $a_0$ is coherent.

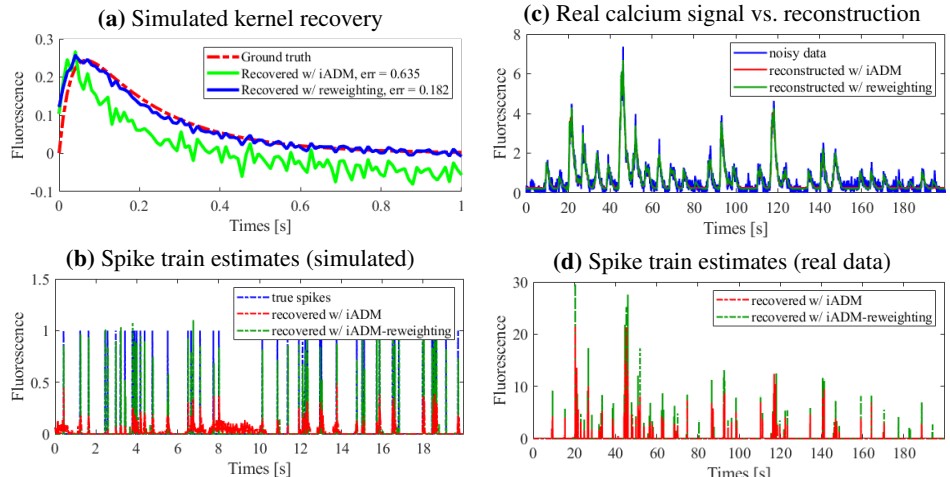

Figure 5: **Deconvolution for calcium imaging** using Algorithm 2 with iADM and with reweighting (Appendix B). *Simulated data:* **(a)** recovered AR2 kernel; **(b)** estimate of spike train. *Real data:* **(c)** reconstructed calcium signal **(d)** estimate of spike train. Reweighting improves estimation quality in each case.

**Recovery performance.** We test recovery probability for varying kernel lengths $p_0$ and sparsity rates $\theta$. To ensure the problem size is sufficiently large, we set $m = 100p_0$. For each $p_0$ and $\theta$, we randomly generate[8] $x \sim_{\mathrm{i.i.d.}} \mathcal{BR}(\theta)$ for both coherent and incoherent $a_0$. We solve ten trials of (3) on clean observation data $a_0 \circledast x_0$ using iADM with $\lambda = \frac{10^{-2}}{\sqrt{p_0\theta}}$. The probability of recovering a signed

---

[8] $\mathcal{BR}(\theta)$ denotes the Bernoulli-Rademacher distribution, which has values $\pm 1$ w.p. $\theta/2$ and zero w.p. $1 - \theta$.

shift of $\boldsymbol{a}_0$ is shown in Figure 4. Recovery is likely when sparsity is low compared to the kernel length. The coherent problem setting has a smaller success region compared to the incoherent setting.

**Momentum and homotopy.** Next, we test the performance of Algorithm 1 with momentum ($\alpha_k = \frac{k-1}{k+2}$; see Pock & Sabach (2016)) and without ($\alpha = 0$). This is done by minimizing $\Psi_{\text{BL}}$ with initialization (5), using clean observations with $p_0 = 10^2$, $m = 10^4$, and $\theta = p_0^{-3/4}$ for coherent and incoherent $\boldsymbol{a}_0$. We also apply homotopy (Algorithm 2) with $\lambda^{(1)} = \max_\ell |\langle s_\ell[\boldsymbol{a}^{(0)}], \boldsymbol{y}\rangle|$ — see Xiao & Zhang (2013), $\lambda^\star = \frac{0.3}{\sqrt{p_0 \lambda}}$, $\eta = 0.8$, and $\delta = 0.1$. The final solve of (3) uses precision $\varepsilon^\star = 10^{-6}$, regardless of method. Figures 4c and 4d show the comparison results on coherent problem settings.

**Comparison to existing methods.** Finally, we compare iADM, and iADM with homotopy, against a number of existing methods for minimizing $\varphi_{\text{BL}}$. The first is *alternating minimization* (Kuo et al., 2019), which at each iteration $k$ minimizes $\boldsymbol{a}^{(k)}$ with $\boldsymbol{x}^{(k)}$ fixed using accelerated (Riemannian) gradient descent with backtracking, and vice versa. The next method is the popular *alternating direction method of multipliers* (Boyd et al., 2011). Finally, we compare against iPALM (Pock & Sabach, 2016) with backtracking, using the unit ball constraint on $\boldsymbol{a}_0$ instead of the unit sphere.

For each method, we deconvolve signals with $p_0 = 50$, $m = 100p_0$, and $\theta = p_0^{-3/4}$ for both coherent and incoherent $\boldsymbol{a}_0$. For both iADM, iADM with homotopy, and iPALM we set $\alpha = 0.3$. For homotopy, we set $\lambda^{(1)} = \max_\ell |\langle s_\ell[\boldsymbol{a}^{(0)}], \boldsymbol{y}\rangle|$, $\lambda^\star = \frac{0.3}{\sqrt{p_0 \lambda}}$, and $\delta = 0.5$. Furthermore we set $\eta = 0.5$ or $\eta = 0.8$ and for ADMM, we set the slack parameter to $\rho = 0.7$ or $\rho = 0.5$ for incoherent and coherent $\boldsymbol{a}_0$ respectively. From Figure 6, we can see that ADMM performs better than iADM in the incoherent case, but becomes less reliable in the coherent case. In both cases, iADM with homotopy is the best performer. Finally, we observe roughly equal performance between iPALM and iADM.

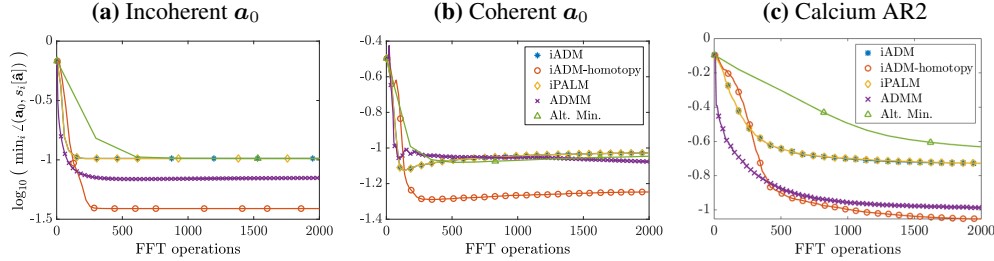

Figure 6: **Algorithmic comparison. (a)** Convergence of various methods minimizing $\Psi_{\text{BL}}$ with incoherent $\boldsymbol{a}_0$ over FFT operations used (for computing convolutions). The y-axis denotes the log of the angle between $\boldsymbol{a}^{(k)}$ and the nearest shift of $\boldsymbol{a}_0$, and each marker denotes five iterations. **(b)** Convergence for coherent $\boldsymbol{a}_0$, and **(c)** with an AR2 kernel for modeling calcium signals.

## 4.2 IMAGING APPLICATIONS

Here we demonstrate the performance and generality of the proposed method. We begin with calcium fluorescence imaging, a popular modality for studying spiking activity in large neuronal populations (Grienberger & Konnerth, 2012), followed by stochastic optical reconstruction microscopy (STORM) (Rust et al., 2006; Huang et al., 2008; 2010), a superresolution technique for *in vivo* microscopy[9].

**Sparse deconvolution of calcium signals.** Neural spike trains created by action potentials, each inducing a transient response in the calcium concentration of the surrounding environment. The aggregate signal can be modeled as a convolution between the transient $\boldsymbol{a}_0$ and the spike train $\boldsymbol{x}_0$. Whilst $\boldsymbol{a}_0$ and $\boldsymbol{x}_0$ both encode valuable information, neither are perfectly known ahead of time.

Here, we first test our method on synthetic data generated using an AR2 model for $\boldsymbol{a}_0$, a shift-coherent kernel that is challenging for deconvolution, see e.g. Friedrich et al. (2017). We set $\boldsymbol{x}_0 \sim_{\text{i.i.d.}}$ Bernoulli($p_0^{-4/5}$) $\in \mathbb{R}^{10^4}$ with additive noise $\boldsymbol{n} \sim_{\text{i.i.d.}} \mathcal{N}(0, 5 \cdot 10^{-2})$. Figures 5a and 5b demonstrate accurate recovery of $\boldsymbol{a}_0$ and $\boldsymbol{x}_0$ in this synthetic setting. Next, we test our method on real data[10]; Figures 5c and 5d demonstrate recovery of spike locations. Although iADM provides

---

[9]Other superresolution methods for microscopy include photoactivated localization microscopy (PALM) (Betzig et al., 2006), and fluorescence photoactivation localization microscopy (fPALM) (Hess et al., 2006).

[10]Obtained at http://spikefinder.codeneuro.org.

decent performance, in the presence of large noise estimation quality can be improved by stronger sparsification methods, such as the reweighting technique by Candes et al. (2008), which we elaborate on in Appendix B. Additionally, Figure 6c shows that the proposed method converges to higher precision in comparison with state-of-the-art methods.

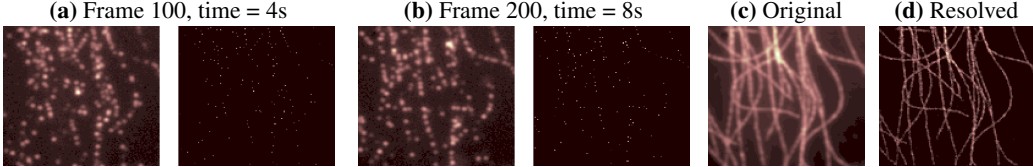

**(a)** Frame 100, time = 4s   **(b)** Frame 200, time = 8s   **(c)** Original   **(d)** Resolved

Figure 7: **SaSD for STORM imaging. (a, b)** Individual frames (left) and predicted point process map using SaSD (right). **(c, d)** shows the original microscopy and the super-resolved image obtained by our method.

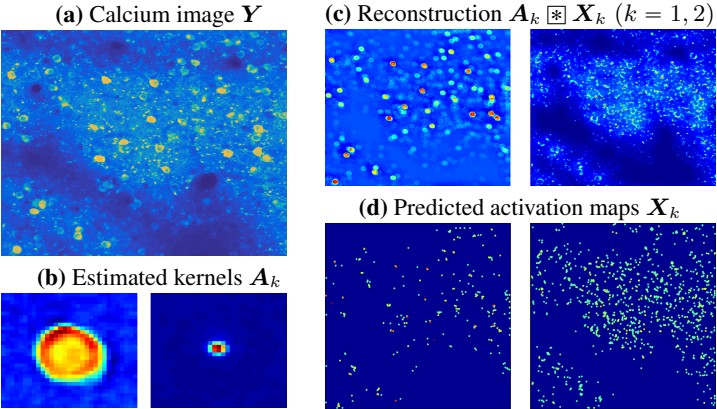

**(a)** Calcium image $Y$   **(c)** Reconstruction $A_k \circledast X_k$ $(k = 1, 2)$

**(b)** Estimated kernels $A_k$   **(d)** Predicted activation maps $X_k$

Figure 8: **Classification of calcium images. (a)** Original calcium image; **(b)** respective kernel estimates; **(c)** reconstructed images with the **(left)** neuron and **(right)** dendrite kernels; **(d)** respective occurence map estimates.

**Super-resolution for fluorescence microscopy.** Fluorescence microscopy is often spatially limited by the diffraction of light; its wavelength (several hundred nanometers) is often larger than typical molecular length-scales in cells, preventing a detailed characterization of subcellular structures. The STORM technique overcomes this resolution limit by using photoswitchable fluorescent probes to multiplex the image into multiple frames, each containing a subset of the molecules present (Figure 7). If the location of these molecules can be precisely determined for each frame, synthesizing all deconvolved frames will produce a super-resolution microscopy image with nanoscale resolution. For each image frame, the localization task can be formulated via the SaS model

$$\underbrace{Y_t}_{\text{STORM frame}} = \underbrace{\iota A_0}_{\text{point spread function}} \circledast \underbrace{X_{0,t}}_{\text{sparse point sources}} + \underbrace{N_t}_{\text{noise}}, \tag{8}$$

where $\circledast$ denotes 2D convolution. Here we will solve this task on the single-molecule localization microscopy (SMLM) benchmarking dataset[11] via SaSD, recovering both the PSF $A_0$ and the point source maps $X_{0,t}$ simultaneously. We apply iADM with reweighting (Appendix B) on frames of size $128 \times 128$ from the video sequence "Tubulin"; each pixel is of $100\text{nm}^2$ resolution[12], the fluorescence wavelength is 690nm, and the framerate is $f = 25\text{Hz}$. Figure 7 shows examples of recovered activation maps, and the aggregated super-resolution image from all 500 frames, accurately predicting the PSF (see Appendix D) and the activation map for each video frame to produce higher resolution microscopy images.

**Localization in calcium images.** Our methods are easily extended to handle superpositions of multiple SaS signals. In calcium imaging, this can potentially be used to track the neurons in video sequences, a challenging task due to (non-) rigid motion, overlapping sources, and irregular

---

[11]Data can be accessed at http://bigwww.epfl.ch/smlm/datasets/index.html.
[12]Here we solve SaSD on the same $128 \times 128$ grid. In practice, the localization problem is solved on a finer grid, so that the resulting resolution can reach $20 - 30$ nm.

background noise Pnevmatikakis et al. (2016); Giovannucci et al. (2019). We consider frames video obtained via the two-photon calcium microscopy dataset from the Allen Institute for Brain Science[13], shown in Figure 8. Each frame contains the cross section of several neurons and dendrites, which have distinct sizes. We model this as the SaS signal $\boldsymbol{Y}_t = \iota\boldsymbol{A}_1 \circledast \boldsymbol{X}_{1,t} + \iota\boldsymbol{A}_2 \circledast \boldsymbol{X}_{2,t}$, where each summand consists of neurons or dendrites exclusively. By extending Algorithm 2 to recover each of the kernels $\boldsymbol{A}_k$ and maps $\boldsymbol{X}_k$, we can solve this *convolutional dictionary learning* (SaS-CDL; see Appendix C) problem which allows us to separate the dendritic and neuronal components from this image for localization of firing activity, etc. As a result, the application of SaS-CDL as a denoising or analysis tool for calcium imaging videos provides a very promising direction for future research.

## 5 DISCUSSION

Many nonconvex inverse problems, such as SaSD, are strongly regulated by their problem symmetries. Understanding this regularity and when or how it breaks down is important for developing effective algorithms. We illustrate this by combining geometric intuition with practical heuristics, motivated by common challenges in real deconvolution, to produce an efficient and general purpose method that performs well on data arising from a range of application areas. Our approach, therefore, can serve as a general baseline for studying and developing extensions to SaSD, such as SaS-CDL (Bristow & Lucey, 2014; Chun & Fessler, 2017; Garcia-Cardona & Wohlberg, 2018), Bayesian approaches (Babacan et al., 2008; Wipf & Zhang, 2014), and hierarchical SaS models (Chen et al., 2013).

ACKNOWLEDGMENTS

This work was funded by NSF 1343282, NSF CCF 1527809, and NSF IIS 1546411. QQ also acknowledges supports from Microsoft PhD fellowship and the Moore-Sloan fellowship. We would like to thank Gongguo Tang, Shuyang Ling, Carlos Fernandez-Granda, Ruoxi Sun, and Liam Paninski for fruitful discussions.

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

## A  APPROXIMATE BILINEAR LASSO OBJECTIVE

Recall from Section 2.2 of the main text that SaSD can be formulated as the Bilinear Lasso problem

$$\min_{\boldsymbol{a}\in\mathbb{S}^{p-1},\boldsymbol{x}\in\mathbb{R}^m} \left[ \Psi_{\mathrm{BL}}(\boldsymbol{a},\boldsymbol{x}) \;\doteq\; \tfrac{1}{2}\|\boldsymbol{y}-\iota\boldsymbol{a}\circledast\boldsymbol{x}\|_2^2 + \lambda\|\boldsymbol{x}\|_1 \right]. \tag{9}$$

Unfortunately, this objective is challenging for analysis. A major culprit is that its marginalization

$$\varphi_{\mathrm{BL}}(\boldsymbol{a}) \;\doteq\; \min_{\boldsymbol{x}}\left\{ \tfrac{1}{2}\|\boldsymbol{y}-\iota\boldsymbol{a}\circledast\boldsymbol{x}\|_2^2 + \lambda\|\boldsymbol{x}\|_1 \right\}, \tag{10}$$

generally does not admit closed form solutions due the convolution with $\boldsymbol{a}$ in the squared error term. This motivates Kuo et al. (2019) to study the nonconvex formulation

$$\min_{\boldsymbol{a}\in\mathbb{S}^{p-1},\boldsymbol{x}\in\mathbb{R}^m} \left[ \Psi_{\mathrm{ABL}}(\boldsymbol{a},\boldsymbol{x}) \;\doteq\; \tfrac{1}{2}\|\boldsymbol{x}\|_2^2 - \langle\iota\boldsymbol{a}\circledast\boldsymbol{x},\boldsymbol{y}\rangle + \|\boldsymbol{y}\|_2^2 + \lambda\|\boldsymbol{x}\|_1 \right]. \tag{11}$$

We refer to (11) as the *Approximate Bilinear Lasso* formulation, and it is quite easy to see that $\Psi_{\mathrm{ABL}}(\boldsymbol{a},\boldsymbol{x}) \approx \Psi_{\mathrm{BL}}(\boldsymbol{a},\boldsymbol{x})$ when $\|\boldsymbol{a}\circledast\boldsymbol{x}\|^2 \approx \|\boldsymbol{x}\|^2$, i.e. if $\boldsymbol{a}$ is shift-incoherent, or $\mu(\boldsymbol{a}) \approx 0$. The marginalized objective function $\varphi_{\mathrm{BL}}(\boldsymbol{a}) \doteq \min_{\boldsymbol{x}}\Psi_{\mathrm{DQ}}(\boldsymbol{a},\boldsymbol{x})$ now has the closed form expression

$$\varphi_{\mathrm{ABL}}(\boldsymbol{a}) \;\doteq\; -\tfrac{1}{2}\|\mathrm{soft}_\lambda[\breve{\boldsymbol{a}}\circledast\boldsymbol{y}]\|_2^2. \tag{12}$$

Here soft denotes the elementwise soft-thresholding operator $\mathrm{soft}_t(x_i) = \mathrm{sign}(x_i)\cdot\max(|x_i|-t,0)$, and $\breve{\boldsymbol{a}}$ denotes the *adjoint kernel* of $\boldsymbol{a}$, i.e. the kernel s.t. $\langle\iota\boldsymbol{a}\circledast\boldsymbol{u},\boldsymbol{v}\rangle = \langle\boldsymbol{u},\breve{\boldsymbol{a}}\circledast\boldsymbol{v}\rangle\ \forall\boldsymbol{u},\boldsymbol{v}\in\mathbb{R}^m$.

### A.1  LANDSCAPE GEOMETRY

The rest of Section 2.2 discusses the regional characterization of $\varphi_{\mathrm{ABL}}$ in the span of a small number of shifts from $\boldsymbol{a}_0$. This language is made precise in the form of the subsphere

$$\mathcal{S}_{\mathcal{I}} \;\doteq\; \left\{ \textstyle\sum_{\ell\in\mathcal{I}} \alpha_\ell s_\ell[\iota\boldsymbol{a}_0] \,:\, \alpha_\ell\in\mathbb{R} \right\} \bigcap \mathbb{S}^{p-1}, \tag{13}$$

spanned by a small set of cyclic shifts of $\iota\boldsymbol{a}_0$. Although we will not discuss the explicit distance function here, the characterization by Kuo et al. (2019) holds whenever $\boldsymbol{a}$ is close enough to such a subsphere with $|\mathcal{I}| \leqslant 4\theta p_0$, where $\theta$ is the probability that any individual entry of $\boldsymbol{x}_0$ is nonzero. Suppose we have $\boldsymbol{a} \approx \sum_{\ell\in\mathcal{I}} \alpha_\ell s_\ell[\iota\boldsymbol{a}_0]$ for some appropriate index set $\mathcal{I}$. Note that if $\mu_s\boldsymbol{a}_0 \approx 0$, then $\mu_s\boldsymbol{a} \approx 0$, $\forall\boldsymbol{a}\in\mathcal{S}_{\mathcal{I}}$. Now let $\alpha_{(1)}$ and $\alpha_{(2)}$ be the first and second largest coordinates of the shifts participating in $\boldsymbol{a}$, and let $s_{(1)}[\boldsymbol{a}_0]$ and $s_{(2)}[\boldsymbol{a}_0]$ be the corresponding shifts. Then

- If $\left|\frac{\alpha_{(2)}}{\alpha_{(1)}}\right| \approx 0$, then $\boldsymbol{a}$ is in a strongly convex region of $\varphi_{\mathrm{ABL}}$, containing a *single* local minimizer corresponding to $s_{(1)}[\boldsymbol{a}_0]$.

- If $\left|\frac{\alpha_{(2)}}{\alpha_{(1)}}\right| \approx 1$, then $\boldsymbol{a}$ is near a saddle-point, with *negative curvature* pointing towards $s_{(1)}[\boldsymbol{a}_0]$ and $s_{(2)}[\boldsymbol{a}_0]$. If $\left|\frac{\alpha_{(3)}}{\alpha_{(2)}}\right| \approx 0$, i.e. $s_{(1)}[\boldsymbol{a}_0]$ and $s_{(2)}[\boldsymbol{a}_0]$ are the only two participating shifts, then $\varphi_{\mathrm{ABL}}$ is also characterized by *positive curvature* in all orthogonal directions.

- Otherwise, $\langle-\mathrm{grad}\,\varphi_{\mathrm{ABL}}(\boldsymbol{a}),\boldsymbol{z}-a\rangle$ takes on a *large positive value*, for either $u = s_{(1)}[\boldsymbol{a}_0]$ or $u = s_{(2)}[\boldsymbol{a}_0]$, i.e. the negative Riemannian gradient is large and points towards one of the participating shifts.

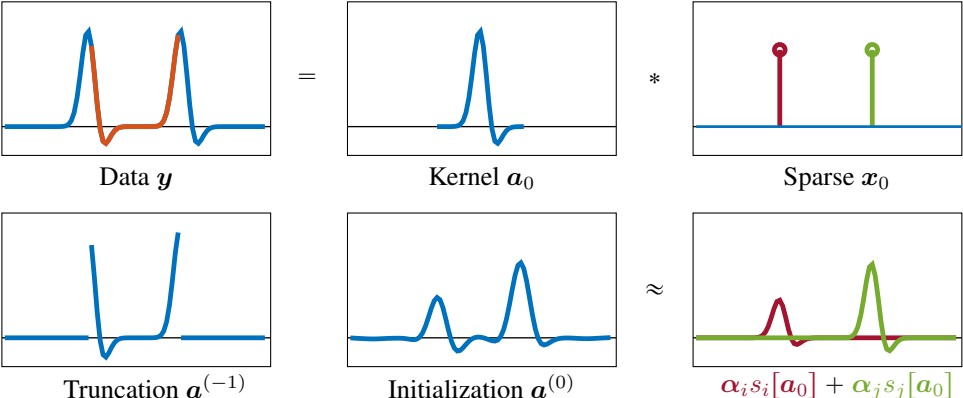

Figure 9: **Data-driven initialization for $a$:** using a piece of the observed data $y$ to generate a good initial point $a^{(0)}$. *Top*: data $y = a_0 \circledast x_0$ is a superposition of shifts of the true kernel $a_0$. *Bottom*: a length-$p_0$ window contains pieces of just a few shifts. *Bottom-center*: one step of the generalized power method approximately fills in the missing pieces, yielding an initialization that is close to a linear combination of shifts of $a_0$ (*right*).

This is an example of a *ridable saddle* property (Jin et al., 2017) that allows many first and second-order methods to locate local minimizers. Since all local minimizers of $\varphi_{\mathrm{ABL}}$ near $\mathcal{S}_{\mathcal{I}}$ must correspond to signed-shifts of $a_0$, this guarantees that the Approximate Bilinear Lasso formulation can be efficiently solved to recover $a_0$ (and subsequently $x_0$) for incoherent $a_0$, as long as $a$ is initialized near some appropriate subsphere and the sparsity coherence tradeoff $p_0\theta \lesssim (\mu_s(a_0))^{-1/2}$ is satisfied. We note that this is a poor tradeoff rate, which reflects that the Approximate Bilinear Lasso formulation is non-practical and cannot handle SaSD problems involving kernels with high shift-coherence.

## A.2 DATA-DRIVEN INITIALIZATION

For the SaS-BD problem, we usually initialize $x$ by $x^{(0)} = 0$, so that our initialization is sparse. For the optimization variable $a \in \mathbb{R}^n$, recall from Section 2.2 in the main text that it is desirable to obtain an initialization $a^0$ which is close to the intersection of $\mathbb{S}^{p-1}$ and a subsphere $\mathcal{S}_{\mathcal{I}}$ spanned by a few shifts of $a_0$. When $x_0$ is sparse, our measurement $y$ is a linear combination of a few shifts of $a_0$. Therefore, an arbitrary consecutive $p_0$-length window $\widetilde{y}_i \doteq [y_i \; y_{i+1} \dots y_{i+p_0-1}]^T$ of the data $y$ should be not far away from such a subspace $\mathcal{S}_{\mathcal{I}}$. As illustrated in Figure 9, one step of the generalized power method (Kuo et al., 2019)

$$\widetilde{a}^{(0)} \doteq \mathcal{P}_{\mathbb{S}^{p-1}}\left(\left[\; \mathbf{0}_{p-1} \; ; \; \widetilde{y}_i \; ; \; \mathbf{0}_{p-1} \; \right]\right) \tag{14}$$

$$a^{(0)} = \mathcal{P}_{\mathbb{S}^{p-1}}\left(-\nabla\varphi_{\mathrm{ABL}}\left(\widetilde{a}^{(0)}\right)\right) \tag{15}$$

produces a refined initialization that is very close to a subspace $\mathcal{S}_{\mathcal{I}}$ spanned by a few shifts of $a_0$ with $|\mathcal{I}| \approx \theta p_0$. However, (15) is a relatively complicated for a simple idea. In practice, we find that the simple initialization $a^{(0)} = \widetilde{a}^{(0)}$ from (14) works suitably well for solving SaSD with (9).

## A.3 COMPARISON TO THE BILINEAR LASSO

Although it is easy to see that $\Psi_{\mathrm{ABL}}(a)$ and $\Psi_{\mathrm{BL}}(a)$ are similar as long as $\mu(a) \approx 0$, it is also clear that these two quantities can be very different when $\mu(a)$ is large. This is especially significant when $\mu(a_0)$ is itself large, as the desired solutions for $a$ are then also coherent.

From Figure 10, we can see that these changes are reflected in the low-dimensional subspheres (13) spanned by adjacent shifts of $a_0$. Compared to the incoherent case, $\varphi_{\mathrm{BL}}$ also takes on small values in regions between adjacent shifts, creating a "global valley" on the subsphere. Theoretically, this makes it difficult to ensure exact recovery of up to symmetry when $a_0$ is coherent, and the objective function becomes much more complicated. This is not a significant issue in terms of practical computation, however, since adjacent shifts of $a_0$ become indistinguishable as $\mu(a_0) \to 1$, meaning that one only needs to ensure that $a$ lands in the "global valley" to achieve good estimates of $a_0$ up to symmetry.

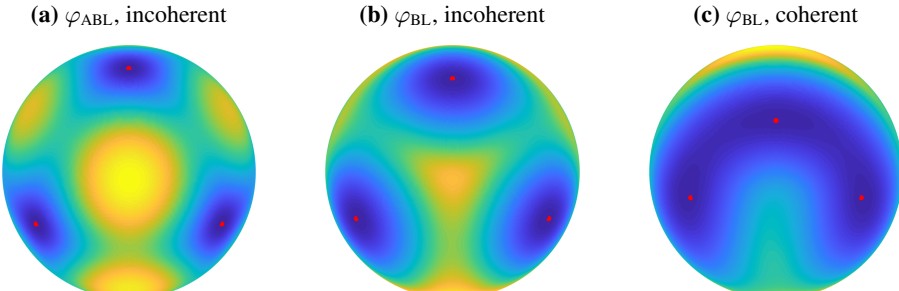

**(a)** $\varphi_{\mathrm{ABL}}$, incoherent  **(b)** $\varphi_{\mathrm{BL}}$, incoherent  **(c)** $\varphi_{\mathrm{BL}}$, coherent

Figure 10: **Low-dimensional subspheres** spanned by shifts of $\boldsymbol{a}_0$. Subfigures **(a,b)** present the optimization landscapes of $\varphi_{\mathrm{ABL}}(\boldsymbol{a})$ and $\varphi_{\mathrm{BL}}(\boldsymbol{a})$, for $\boldsymbol{a} \in \mathbb{S}^{p-1} \bigcap \mathrm{span}\{\boldsymbol{a}_0, s_1[\boldsymbol{a}_0], s_2[\boldsymbol{a}_0]\}$, with higher values being brighter. The red dots denote the shifts of $\boldsymbol{a}_0$. Subfigure **(c)** shows the landscape $\varphi_{\mathrm{BL}}$ when $\boldsymbol{a}_0$ is coherent, which significantly departs from the landscapes of **(a,b)**, but still retains symmetry breaking curvature.

## B  REWEIGHTED SPARSE PENALIZATION

When $\boldsymbol{a}_0$ is shift-coherent, minimization of the objective $\Psi_{\mathrm{BL}}$ with respect to $\boldsymbol{x}$ becomes sensitive to perturbations, creating "smudging" effects on the recovered map $\boldsymbol{x}$. These resolution issues can be remedied with stronger *concave* regularizers. A simple way of facilitating this with the Bilinear Lasso is to use a reweighting technique (Candes et al., 2008). The basic idea is to adaptively adjust the penalty by considering a weighted variant of the original Bilinear Lasso problem from (9),

$$\min_{\boldsymbol{a} \in \mathbb{S}^{p-1}, \boldsymbol{x} \in \mathbb{R}^m} \Psi_{\mathrm{BL}}^{\boldsymbol{w}}(\boldsymbol{a}, \boldsymbol{x}) \doteq \tfrac{1}{2} \|\boldsymbol{y} - \boldsymbol{a} \circledast \boldsymbol{x}\|_2^2 + \lambda \|\boldsymbol{w} \odot \boldsymbol{x}\|_1 \tag{16}$$

where $\boldsymbol{w} \in \mathbb{R}_+^m$ and $\odot$ denotes the Hadamard product. Here we will set the weights $\boldsymbol{w}$ to be roughly inverse to the magnitude of the true signal $\boldsymbol{x}_0$, i.e.,

$$w_i = \frac{1}{|x_{0,i}| + \varepsilon}. \tag{17}$$

---

**Algorithm 3** Reweighted Bilinear Lasso

**Input:**  Initializations $\hat{\boldsymbol{a}}^{(0)}, \hat{\boldsymbol{x}}^{(0)}$, penalty $\lambda > 0$
**Output:**  Local minimizers $\hat{\boldsymbol{a}}^{(j)}, \hat{\boldsymbol{x}}^{(j)}$ of $\Psi_{\mathrm{BL}}^{\boldsymbol{w}^{(j)}}$.
  Initialize $\boldsymbol{w}^{(1)} = \boldsymbol{1}_m, j \leftarrow 1$.
  **while** not converged **do**
    Using the initialization $\left(\hat{\boldsymbol{a}}^{(j-1)}, \hat{\boldsymbol{x}}^{(j-1)}\right)$ and weight $\boldsymbol{w}^{(j)}$, solve (16) — e.g. with iADM — to obtain solution $\left(\hat{\boldsymbol{a}}^{(j)}, \hat{\boldsymbol{x}}^{(j)}\right)$;
    Set $\varepsilon$ with (19) and update the weights as

$$\boldsymbol{w}^{(j+1)} = \frac{1}{\left|\hat{\boldsymbol{x}}^{(j)}\right| + \varepsilon}. \tag{18}$$

    Update $\ell \leftarrow \ell + 1$.
  **end while**

---

In addition to choosing $\lambda > 0$, here $\varepsilon > 0$ trades off between sparsification strength (small $\varepsilon$) and algorithmic stability (large $\varepsilon$). Let $|x|_{(i)}$ denote the $i$-th largest entry of $|\boldsymbol{x}|$. For experiments in the main text, we set

$$\varepsilon = \max \left\{ |x|_{(\lceil n/\log(m/n) \rceil)}, 10^{-3} \right\}. \tag{19}$$

Starting with the initial weights $\boldsymbol{w}^{(0)} = \boldsymbol{1}_m$, Algorithm 3 successively solves (16), updating the weights using (17) at each outer loop iteration $j$. As $j \to \infty$, this method becomes equivalent to replacing the $\ell_1$-norm in (9) with the nonconvex penalty $\sum_i \log(|x_i| + \varepsilon)$ (Candes et al., 2008).

We can easily adopt our iADM algorithm to solve this subproblem, by taking the proximal gradient on $x$ with a different penalty $\lambda_i$ for each entry $x_i$. Figure 11, as well as calcium imaging experiments in Section 4.2, Figure 5 of the main text, demonstrate improved estimation as a result of this method.

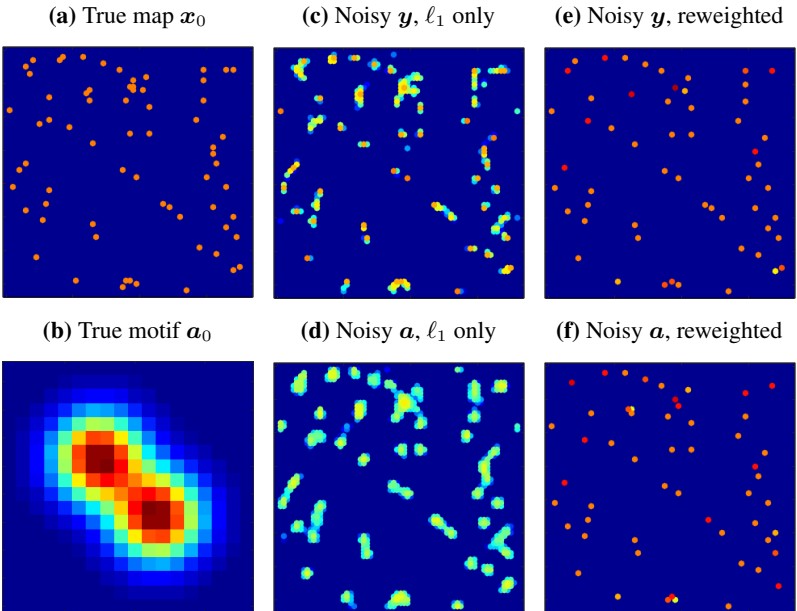

**(a)** True map $x_0$  **(c)** Noisy $y$, $\ell_1$ only  **(e)** Noisy $y$, reweighted

**(b)** True motif $a_0$  **(d)** Noisy $a$, $\ell_1$ only  **(f)** Noisy $a$, reweighted

Figure 11: **Recovery of $x_0$ with $\ell_1$-reweighting. (a, b)** Truth signals. **(c)** Solving $\min_x \Psi_{\mathrm{BL}}(a, x)$ with noisy data and coherent $a_0$ leads to low-quality estimates of $x$; **(d)** performance suffers further when $a$ is a noisy estimate of $a_0$. **(e, f)** Reweighted $\ell_1$ minimization alleviates this issue significantly.

## C EXTENSION FOR CONVOLUTIONAL DICTIONARY LEARNING

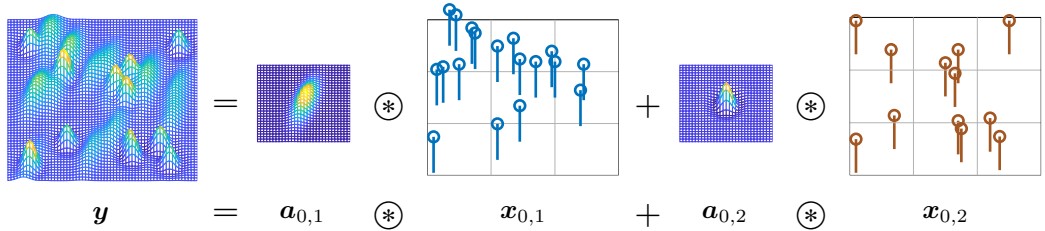

$$y \quad = \quad a_{0,1} \quad \circledast \quad x_{0,1} \quad + \quad a_{0,2} \quad \circledast \quad x_{0,2}$$

Figure 12: **Convolutional dictionary learning.** Simultaneous recovery for multiple unknown kernels $\{a_{0,k}\}_{k=1}^N$ and sparse activation maps $\{x_{0,k}\}_{k=1}^N$ from $y = \sum_{k=1}^N a_{0,k} \circledast x_{0,k}$.

**(a)** PSF in 2D    **(b)** PSF in 3D

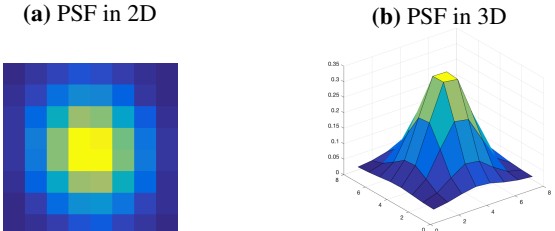

Figure 13: **Estimated PSF for STORM imaging.** The left hand side shows the estimated $8 \times 8$ PSF in 2D, the right hand side visualizes the PSF in 3D.

The optimization methods we introduced for SaSD here can be naturally extended for sparse blind deconvolution problems with multiple kernels/motifs (a.k.a. convolutional dictionary learning; see Garcia-Cardona & Wohlberg (2018)), which have broad applications in microscopy data analysis (Yellin et al., 2017; Zhou et al., 2014; Cheung et al., 2018) and neural spike sorting (Ekanadham et al., 2011; Rey et al., 2015; Song et al., 2018). As illustrated in Figure 12, the new observation $y$ is

the sum of $N$ convolutions between short kernels $\{\boldsymbol{a}_{0,k}\}_{k=1}^N$ and sparse maps $\{\boldsymbol{x}_{0,k}\}_{k=1}^N$,

$$\boldsymbol{y} = \sum_{k=1}^N \iota\boldsymbol{a}_{0,k} \circledast \boldsymbol{x}_{0,k}, \qquad \boldsymbol{a}_{0,k} \in \mathbb{R}^{p_0}, \quad \boldsymbol{x}_{0,k} \in \mathbb{R}^m, \quad (1 \leqslant k \leqslant N). \tag{20}$$

The natural extension of SaSD, then, is to recover $\{\boldsymbol{a}_{0,k}\}_{k=1}^N$ and $\{\boldsymbol{x}_{0,k}\}_{k=1}^N$ up to signed, shift, and permutation ambiguities, leading to the SaS convolutional dictionary learning (SaS-CDL) problem. The SaSD problem can be seen as a special case of SaS-CDL with $N = 1$. Based on the Bilinear Lasso formulation in (9) for solving SaSD, we constrain all kernels $\boldsymbol{a}_{0,k}$ over the sphere, and consider the following nonconvex objective:

$$\min_{\{\boldsymbol{a}_k\}_{k=1}^N, \, \{\boldsymbol{x}_k\}_{k=1}^N} \frac{1}{2} \left\| \boldsymbol{y} - \sum_{k=1}^N \boldsymbol{a}_k \circledast \boldsymbol{x}_k \right\|_2^2 + \lambda \sum_{k=1}^N \|\boldsymbol{x}_k\|_1, \quad \text{s.t.} \quad \boldsymbol{a}_k \in \mathbb{S}^{p-1} \quad (1 \leqslant k \leqslant N). \tag{21}$$

Similar to the idea of solving the Bilinear Lasso in (9), we optimize (21) via iADM, by taking alternating descent steps on $\{\boldsymbol{a}_k\}_{k=1}^N$ and $\{\boldsymbol{x}_k\}_{k=1}^N$ with the other variable fixed.

## D    SUPER-RESOLUTION WITH STORM IMAGING

For point source localization in STORM frames, recall that we use the SaS model from Section 4.2.2,

$$\underbrace{\boldsymbol{Y}_t}_{\text{STORM frame}} = \underbrace{\iota\boldsymbol{A}_0}_{\text{point spread function}} \boxasterisk \underbrace{\boldsymbol{X}_{0,t}}_{\text{sparse point sources}} + \underbrace{\boldsymbol{N}_t}_{\text{noise}}. \tag{22}$$

We then apply our SaSD method to recover both $\boldsymbol{A}_0$ and $\boldsymbol{X}_{0,t}$ from $\boldsymbol{Y}_t$. We show our recovery of $\boldsymbol{X}_{0,t}$ as well as the super-resolved image using all available frames in Figure 6 of the main text. Since the main objective of STORM imaging is to recover the point sources, we have deferred the recovered PSF $\boldsymbol{A}_0$ to Figure 13 here.

