# OpenReview forum: "Short and Sparse Deconvolution --- A Geometric Approach"
_ICLR.cc/2020/Conference — Accept (Poster)_

### Official Review · AnonReviewer1 · 2019-10-23
**Official Blind Review #1**

**Rating:** 6

**Review:**

* Summary

The work considers sparse and short blind deconvolution problem, which is to inverse a convolution of a sparse source (such as spikes at cell locations in microscopy) with a short (of limited spatial size) kernel or point spread function, not known in advance. This is posed as a bilinear lasso optimization problem. The work applies a non-linear optimization method with some practical improvements (such as data-driven initialization, momentum, homotopy continuation) and applies it to 3 real imagine problems.

I think the work can be described as bridging from the known problem formulation, available theoretical understanding of its properties and available selection of optimization methods, to an implementation that can be applied in practical cases. The practical improvements made are not specifically novel, but result in a well-fit optimization method.
On the practical end, it is shown that the method can be applied in multiple cases but it is not demonstrated to give practical improvements over any alternative reconstruction approaches. The emphasis is more on showcasing possible further extensions (many deferred to appendices). I view this work as not very strong but a valid contribution.

* Detailed Comments

While the ideas from a provable algorithm by Kuo et al. are used here (sphere constraints, data-driven initialization), can the authors show in their setting, this leads as well to optimal recovery guarantees for sufficiently simple problems?

Somehow the discussion advocates spherical constraint, because different shifts of the kernel become different local minima. But at the same time, this increases the non-linearity of the problem, and thus makes it more difficult to solve. Although, these multiple local optima provide equivalent solutions, I find this somewhat counter-intuitive. Cannot the shift ambiguity be resolved in a convex manner, e.g. by fixing the mean value of the kernel?

In Fig 1 an explanation of the axis and projections would be needed.
While in the introduction, the setting m>>p0 is assumed, in the experiments n0 is used to denote kernel width and some experiments actually work in the setting of comparable values such as n0=50, m=100.

What are similarities / differences to related reconstruction problems such as non-negative matrix factorization with sparseness constraints?



**Experience Assessment:**

I do not know much about this area.

**Review Assessment: Checking Correctness Of Derivations And Theory:**

I assessed the sensibility of the derivations and theory.

**Review Assessment: Checking Correctness Of Experiments:**

I assessed the sensibility of the experiments.

**Review Assessment: Thoroughness In Paper Reading:**

I read the paper at least twice and used my best judgement in assessing the paper.

---

> ### Author Response · Authors · 2019-11-15
> **Reply to Reviewer #1**
>
> We thank the reviewer for the accurate interpretation and the appreciation of our work. We address your comments below.
>
> *Comparison with state-of-the-art methods on practical problems.  As suggested by the reviewer, we have added an experiment that compares the proposed method with state-of-the-art algorithms (in Figure 6a&b) on the task of sparse deconvolution of calcium signals (see Figure 6c).
>
> *Comments on spherical constraints. The sphere constraint removes the scaling ambiguity discussed in Section 2. We agree with the reviewer that it is seemingly counter-intuitively that the spherical constraint is a good choice here. But our findings do suggest that, rather than making the problem more complicated (as the comment suggests), the spherical constraint actually plays an important role in making the SaSD problem easy to solve! Negative curvature in symmetry breaking directions (see Figure 1), the key geometric feature of the landscape, results from interactions between the (Euclidean) curvature of the objective function and the curvature of the sphere.
>
> In this setting optimizing over a ball is equivalent to optimizing over the sphere, since minimizers over the ball always occur at the boundary. Finally, similar to the question of Reviewer 2 above, convex constraints which remove the shift ambiguity tend to create spurious local minimizers. The constraint forces us to find one particular shift, if our initialization is close to other shifts, causing local methods to become trapped. In a sense, the presence of multiple global minimizers helps: although we do not know which one we will find, in the theoretical setting of Kuo et. al, we know we will find one of them.
>
> *Similarities/differences to other nonconvex problems. Indeed, similar geometric structures appear in other nonconvex problems such as sparse dictionary learning, which exhibits permutation symmetry, preventing natural convex relaxations. However, for problems like (sparse) nonnegative matrix factorization, as the nonnegative orthant often creates spurious local minimizers on the boundary of the constraints, our intuitions here cannot be carried through to that problem yet.
>
> *Minor comments. For Figure 1 (a-c), we have added a comment in the caption that the upper denotes the function value in height, while the bottom plots the function value over the sphere. We thank the reviewer for catching an inconsistency in the statements of $p_0$ and $m$ between the introduction and experiments. Indeed, our statement of the experiment part that $m=100$ on Page 6 is incorrect, it should be $m = 100 p_0$, which has been revised.

---

### Official Review · AnonReviewer2 · 2019-10-30
**Official Blind Review #2**

**Rating:** 3

**Review:**

This paper analyzes some of the optimization challenges presented by a particular formulation of "short-and-sparse deconvolution" and proposes a new general purpose algorithm to try to alleviate them. The paper is reasonably clearly written and the experimental results seem impressive (as a non-specialist in this area). However the experimental investigation on real data does not compare to a baseline, which seems like an essential part of investigating the proposed method. If this were corrected I would recommend an accept.

I would like to make it clear that I have little experience in this area and am not familiar with relevant previous literature, so I was only able to roughly judge novelty of the ideas and how the experimental results might compare with existing approaches.


Major comments:

For the experimental investigations on real data there was no baseline presented. Are there no task-specific algorithms that have previously been developed for deconvolution of calcium signals, fluoresence microscopy and calcium imaging? At the very least it would be helpful to compare to one of the other general purpose methods used for figure 6.

It seemed like a lot of the paper is taken up with a recap of the results presented in Kuo et al. (2019), and it didn't always seem clear exactly what the relevance of these results were to the present paper. In particular it seemed strange to me to devote so much space in section 2 to the ABL objective given that it is not used (as far as I can tell) for the proposed algorithm.


Minor comments:

The abstract says "We leverage... sphere constraints, data-driven initialization". Is the effect of these actually investigated experimentally?

In the abstract, "This is used to derive a provable algorithm" makes it sounds to me like this will be done in the current paper.

In the abstract, a reference for the "due to the spectral decay of the kernel a_0" claim would be helpful.

In the notation section, the definition of the Riemannian gradient seems a little sloppy mathematically: strictly f needs to be defined on $\reals^p$ (or an open subset of $\reals^p$ containing $S^{p - 1}$) in order for the right side of the gradient expression to be defined.

At the start of section 2, it would be helpful to give a one-sentence description of the unifying theme of the section.

It might be helpful to explicitly state that "coherence" is related to the strength of temporal / spatial correlations to aid developing the reader's intuition for the meaning of $\mu(a)$.

Throughout the paper the problems that multiple equivalent local optima (due to shift symmetry) cause come up repeatedly. What happens if this symmetry is removed straightforwardly by adding a term to the cost function to encourage a particular value of $a$ to be the largest?

In section 2.1, for "It analyzes an ABL...", it's not completely clear what "It" refers to (I presume Kuo et. al (2019) from context).

Marginalization in my experience refers to a "partial summing" operation, whereas just above (4) it is used to refer to a "partial maximization" operation. This seems non-standard to me, but is this usage standard in this field?

I didn't understand the relevance of the sentence "Under its marginalization... smaller dimension p << m." to the present paper. The sentence also seemed vague and difficult to understand if you were not already familiar with this result. It should also have a reference to justify this claim.

It wasn't clear to me whether figure 1 were schematic "rough intuition" diagrams intended merely to be suggestive, or provably showing the type of behavior that happens in all cases. Also, what are the axes, generic "parameter space", I guess? I also didn't follow why (a) appears projected on to a plane while (b) and (c) appear projected on to a sphere(?)

In section 3, under "momentum acceleration", it would be helpful to justify the claim that "In shift-coherent settings, the Hessian... ill-conditioned...".

In figure 4, the axis labels are much too small to read. In figure 5 (b), the red poorer results obscure the green better results.

In figure 4 (d), is it fair to say that the main convergence speed improvement for homotopy-iADM is in going faster from "quite near" the optimum to "really near" it, rather than getting "quite near" it in the first place? If so, isn't the latter more often what's relevant in practical applications?

In figure 5, it's a little confusing to switch color meanings between (a) and (b).

Reweighting seems to have a dramatic positive effect in figure 5. Is it worth investigating its effect in the other experiments as well, for example in figure 4?

In figure 6 (b), is there any reason not to compare against standard black box optimizers like vanilla SGD, ADAM, etc (possibly with projection to satisfy the sphere constraint if necessary)? Would they perform very badly?

Typo "Whilst $a_0$ nor $x_0$" should be "Whilst $a_0$ and $x_0$".

What does the square-boxed convolution operator in (8) mean?


**Experience Assessment:**

I do not know much about this area.

**Review Assessment: Checking Correctness Of Derivations And Theory:**

I assessed the sensibility of the derivations and theory.

**Review Assessment: Checking Correctness Of Experiments:**

I assessed the sensibility of the experiments.

**Review Assessment: Thoroughness In Paper Reading:**

I read the paper thoroughly.

---

> ### Author Response · Authors · 2019-11-15
> **Reply to Reviewer #2 (Major Comments)**
>
> We thank the reviewer for the detailed review and constructive comments.
>
> *Comparison on practical problems. As suggested by the reviewer, we have added an experiment that compares the proposed method with state-of-the-art algorithms (in Figure 6a&b) on the task of sparse deconvolution of calcium signals under AR2 model (see Figure 6c).
>
> *Relation to the result of Kuo et al. (2019). The theory in Kuo et al. is based on an approximation $\Psi_{\text{ABL}} $ to the Bilinear Lasso $\Psi_{\text{BL}} $ (see Appendix A). These two functions have the same scale-shift symmetry. On incoherent problems (see definition in (2)), they exhibit similar critical points: minimizers are signed shifts of the ground truth, saddle points occur near superpositions of shifts, with negative curvature in symmetry breaking directions (compare, e.g., the left and center panels of Figure 10 of Appendix A of the submission).
>
> Despite this similarity, $\Psi_{\text{BL}} $ and $\Psi_{\text{ABL}} $ are not identical, especially for the type of coherent problems encountered in applications in imaging and the sciences (right panel of Figure 10). The goal of the paper is to leverage intuitions from Kuo et al. (derived under incoherence assumptions) to build practical methods that can solve these highly coherent problems. Of course, the theory of Kuo et al. does not have direct mathematical implications for Bilinear Lasso or for highly coherent problems. There are a number of roles that theory can play in the computational sciences. The most direct role is in clarifying fundamental limits and providing performance guarantees. Another potential role is in providing ways of thinking: models, pictures, intuitions, which can have more practical value than the mathematical theorems themselves.

---

> > ### Author Response · Authors · 2019-11-15
> > **Reply to Reviewer #2 (Minor Comments)**
> >
> > We revised the paper according to the reviewer's minor comments. We address several important aspects as follows.
> >
> > *Data-driven initializations. We have experimentally tested our methods with random initialization vs. data-driven initialization. We observe that methods with random initialization sometimes converge to a spurious local minimizer (e.g., shift truncation) instead of the target solution. In contrast, methods using data-driven initialization always converge to the target solution. Due to space limitations, the results are not included here.
> >
> > *Spherical constraints. We refer the reviewer to our reply to Reviewer 1.
> >
> > *Explanation of the definition of coherence $\mu(\mathbf a_0)$ in (2). The coherence parameter $\mu(\mathbf a_0)$ characterizes the largest similarity between $\mathbf a_0$ and all its shifts. Intuitively, larger $\mu(\mathbf a_0)$ implies that $a_0$ is closer to one of its shifts, making the nonconvex landscape less favorable for optimization (see Figure 10 on Page 13). This implies that the SaSD is harder to solve when $\mu(\mathbf a_0)$ becomes larger. We have made this clear in the revision.
> >
> > *Breaking symmetry with additional constraints. Adding constraints to break symmetry is a natural idea. Unfortunately, symmetry breaking constraints tend to introduce spurious local minimizers. Since the constraint forces us to find one particular shift, if the initialization is close to other shifts, local methods can easily get trapped. Theory indicates that, surprisingly, there is no need to break symmetry: although we cannot predict ahead of time which shift will be recovered, we will recover some shift.
> >
> > *Riemannian gradient and notations on top of Page 2. Although the function is defined on the sphere, the Riemannian gradient is defined in the tangent space. It measures the slope of the function over the sphere (rather than over the ambient space). We refer the reviewer to https://www.manopt.org/ for more details.
> >
> > *Explanation of Figure 1. For Figure 1 (a-c), we have added a comment in the caption that the upper denotes the function value in height, while the bottom plots the function value over the sphere for all (a-c).
> >
> > *Explanation of Ill-conditioned Hessian in Section 3. The reason we claim that the Hessian for the coherent kernel $\mathbf a_0$ is ill-conditioned is because the circulant matrix $\mathbf C_{\mathbf a_0}$ of $\mathbf a_0$ is ill-conditioned. We have added a footnote explaining this in the revised version.
> >
> > *Explanation of Figure 4(d). The comparison in the original version is based on function values, which is not fair for homotopy methods using a large $\lambda$ in the beginning. In the revision, we have changed the comparison of the distance between iterate and the ground truth, which we believe is a more fair comparison. As shown in the revised Figure 4(d), the homotopy method has a clear advantage (the spiky jump is due to the adaptation of the values of $\lambda$).
> >
> > *Explanation of partial minimization around (4). We agree with the reviewer that the meaning of sentences below (4) is confusing. We have rephrased them in the revision.
> >
> > *Comparison with SGD, ADAM. As we consider the problem with a single long sequence measurement $\mb y$, we have not compared with methods using mini-batches.
> >
> > *Reweighting Method. In the experiment, we found the reweighting method (see Appendix B) shows superior performance for problems with large noises, by adaptively updating the penalty $\lambda$. We have revised the statement mentioning this phenomenon in Section 4.2. Since the data generation in Figure 4 does not involve any noise, we have not included the reweighting method in the comparison.
> >
> > *Squared-boxed convolution in (8). The square-boxed convolution operator in (8) means 2D convolution, which we want to distinguish from 1D convolution. We have made this clear in the revision.

---

### Decision · Program_Chairs · 2019-12-19

**Decision:**

Accept (Poster)

**Comment:**

The work considers sparse and short blind deconvolution problem, which is to inverse a convolution of a sparse source (such as spikes at cell locations in microscopy) with a short (of limited spatial size) kernel or point spread function, not known in advance. This is posed as a bilinear lasso optimization problem. The work applies a non-linear optimization method with some practical improvements (such as data-driven initialization, momentum, homotopy continuation).

The paper extends the work by Kuo et al. (2019) by providing a practical algorithm for solving those inverse problems. A focus of the paper is to solve the bilinear lasso instead of the approximate bilinear lasso, because this approximation is poor for coherent problems. Having read the rebuttal and the paper, I believe the authors addressed the issues raised by Reviewer #2 in a sufficient way.

small things:
- it would be good to define $\iota$ (zero-padding operator) in (1)
- it would be good to define $p, p_0$ just below (3). They seem to be appearing out of the blue without any direct relation to anything mentioned prior in section 2.
- it would be good to cite some older/historic references for various optimization methods , e.g. [1] below.


[1] Richter & deCarlo
Continuation methods: Theory and applications
IEEE Transactions on Systems, Man, and Cybernetics, 1983
https://ieeexplore.ieee.org/abstract/document/6313131